# Autosomal Recessive Non-Syndromic Deafness: Is AAV Gene Therapy a Real Chance?

**Davide Brotto** [1,2,†], **Marco Greggio** [1,2,*,†], **Cosimo De Filippis** [1] **and Patrizia Trevisi** [1,2]

1    Department of Neuroscience DNS, Otolaryngology Section, Padova University, 35128 Padova, Italy; davide.brotto@unipd.it (D.B.); cosimo.defilippis@unipd.it (C.D.F.); patrizia.trevisi@unipd.it (P.T.)
2    Otolaryngology Unit, Azienda Ospedale Università Padova, 35128 Padova, Italy
*    Correspondence: marcogreg20@gmail.com or marco.greggio.6@studenti.unipd.it
†    These authors contributed equally to this work.

**Abstract:** The etiology of sensorineural hearing loss is heavily influenced by genetic mutations, with approximately 80% of cases attributed to genetic causes and only 20% to environmental factors. Over 100 non-syndromic deafness genes have been identified in humans thus far. In non-syndromic sensorineural hearing impairment, around 75–85% of cases follow an autosomal recessive inheritance pattern. In recent years, groundbreaking advancements in molecular gene therapy for inner-ear disorders have shown promising results. Experimental studies have demonstrated improvements in hearing following a single local injection of adeno-associated virus-derived vectors carrying an additional normal gene or using ribozymes to modify the genome. These pioneering approaches have opened new possibilities for potential therapeutic interventions. Following the PRISMA criteria, we summarized the AAV gene therapy experiments showing hearing improvement in the preclinical phases of development in different animal models of DFNB deafness and the AAV gene therapy programs currently in clinical phases targeting autosomal recessive non syndromic hearing loss. A total of 17 preclinical studies and 3 clinical studies were found and listed. Despite the hurdles, there have been significant breakthroughs in the path of HL gene therapy, holding great potential for providing patients with novel and effective treatment.

**Keywords:** gene therapies; adeno-associated virus (AAV); sensorineural hearing loss; recessive mutations; DFNB; hearing restoration





## 1. Introduction

The occurrence of sensorineural hearing loss in healthy full-term neonates is estimated to be between 1 and 3 per 1000 live births and genetic factors play a critical role in at least 50% of the cases [1]. Hearing loss due to autosomal recessive transmission and autosomal dominant transmission occurs in approximately 80–90% and 10–20% of cases, respectively, while X-linked inheritance is observed in the remaining cases [2]. So far, a total of 61 distinct genes have been documented to carry biallelic mutations associated with autosomal recessive non-syndromic hearing loss (ARNSHL) [3]. Among them, the most frequently involved are GJB2 and GJB6 mutations encoding for connexin 26 and 30 (the most prevalent form of congenital hearing loss), OTOF mutations encoding for otoferlin, and other genetic mutations affecting the STRC, TMC1, TMPRSS3, ILDR1, CAPB2, SYNE4, TMHS and MSRB3 genes. All these mutations can cause sensorineural hearing loss with different hearing configurations and a poorly predictable prognosis regarding hearing, at least at the moment. Nowadays, the only option able to mitigate the consequent deficit involves rehabilitating the auditory function by means of hearing aids or cochlear implantation (for severe to profound hearing loss with no benefit with hearing aids).

Recent research has highlighted the complexity of genetic hearing loss, and it is trying to develop new non-prosthetic approaches for these issues. For this reason, both non-viral and viral delivery therapeutic approaches are currently being developed [4].

Gene replacement is a promising therapeutic approach for hereditary hearing loss, in which viral vectors are used to deliver functional cDNA to "replace" defective genes in dysfunctional cells of the inner ear. Proof-of-concept studies have successfully used this approach to improve auditory function in mice models of hereditary hearing loss [5]. The inner ear can be a good candidate for gene therapy for several reasons: (1) the cochlea is easily accessible by means of minimally invasive surgery; (2) it is reasonably self-contained anatomically, allowing the easy and direct delivery of gene therapy; (3) the local application is performed within a relatively immune-protected environment; and (4) the organ is filled with fluid, thus allowing for the widespread diffusion of the delivered gene.

Animal studies have shown that recombinant adeno-associated virus (rAAV)-mediated gene therapy can at least partially restore inherited deafness caused by gene mutations or deletions [6–11]. The initial characterization of natural AAV serotypes revealed a relatively low transduction rate of inner ear cell types and of outer hair cells [12]. However, recently developed synthetic AAV capsids seem to have overcome this hurdle; AAV9-PHP.B, for example, has been shown to transduce both inner and outer hair cells at high rates in mice and non-human primates [13–15]. As a clinically therapeutic "star vector", AAV seems to be promising for application in clinical research into deafness treatment.

The aim of the present review is to report and summarize what is currently known and ongoing concerning preclinical and clinical AAV gene therapy studies targeting autosomal recessive gene mutations leading to non-syndromic forms of sensorineural hearing impairment.

## 2. Material and Methods

The literature search was performed by two authors independently (MG and DB), according to the PRISMA criteria. To identify potentially relevant documents, the Google Scholar and PubMed bibliographic databases and Clinicaltrials.gov register were searched from January 2023 to September 2023. The scientific literature was searched in PubMed and Google Scholar for preclinical studies and in Clinicaltrials.gov for clinical studies; combinations of terms such as "gene therapy" AND "sensorineural hearing loss" or "genetic hearing loss" or "genetic deafness" and "DFNB" were used. Studies meeting the following inclusion criteria were included: studies published between January 2010 and June 2023; studies written in English; and studies involving both the animal (preclinical) and human (clinical) phase of development and related to experimental gene therapies targeting cochlear pathologies caused by genetic mutations. Only mutations that were autosomal recessively inherited and causative of non-syndromic forms of SNHL were included; in particular, we focused on genetic mutations affecting genes such as GJB2, OTOF, STRC, TMC1, TMPRSS3, ILDR1, CAPB2, SYNE4, TMHS and others. Among the preclinical studies, only gene therapy programs showing the restoration of hearing in DFNB animal models were included. All the records that did not meet these inclusion criteria were excluded. The studies included are reported in two summary tables divided between those in the preclinical phase and those in the clinical phase. In Table 1, which shows the preclinical studies, the animal model, the DFNB targeted, the AAV treatment reagent and the publication year are listed; in Table 2, the specific mutation and DFNB targeted, the AAV treatment reagent, the current clinical phase of development and the clinicatrials.gov identifier (NCT number) are listed.

## 3. Results

A total of 831 records were identified, 827 from databases (823 from Google Scholar, 4 from PubMed) and 4 from the Clinicaltrials.gov register. In the identification process, one from the PubMed database was removed due to duplication. In total, 826 records from the databases (823 from Google Scholar, 3 from PubMed) and 4 from Clinicaltrials.gov were screened: 260 (259 google scholar, 1 PubMed) records from the databases were removed because they were not written in English or were out of the publication year range; no records from Clinicaltrials.gov were removed. A total of 570 reports were sought for retrieval (564 from Google Scholar and 2 from PubMed databases; 4 from Clinicaltrials.gov):

540 reports from the databases (538 Google Scholar, 2 PubMed) were removed because they did not present preclinical or clinical studies advancing the AAV gene therapy approach, because they were related to other topics or to different fields of medicine, or because they were not peer-reviewed papers but theses, medical events, or patent applications (538 from Google Scholar, 2 from PubMed); one report from Crinicaltrials.gov was removed because, although it presented a clinical study testing a form of gene therapy for OTOF gene hearing impairment, the specific gene vector administered was not reported.

A total of 29 reports were assessed for eligibility. Among the reports excluded, nine studies may have appeared to meet the inclusion criteria as they related to gene therapies for the genetic form of sensorineural hearing loss However, (1) the study by Solanes P. et al. reporting that the delivery of CRISPR/Cas9 using AAV-PHP.B in mutated Tmc1 Beethoven mice protects auditory function was excluded because it targeted the autosomal dominant (DFNA36) and not the recessive (DFNB7/11) form of non-syndromic sensorineural hearing loss [16]. (2) For the same reason, the studies by Yoshimura et al. and (3) Iwasa et al. reporting the prevention of progressive hearing loss in a mature murine model of human TMC1 deafness and the rescue of hearing in a TMC-1 related hearing loss mouse model through a mutation-agnostic RNA interference with engineered replacement were excluded [17,18]. (4) The clinical study by Fujioka M. et al. targeting DFNB4 deafness was excluded because it targets a syndromic form of SNHL and also does not deal with a gene therapy approach [19]. (5) The study by Guo et al. adopted a form of AAV gene therapy in a GJB2 animal model; however, after generating mice with the inducible Sox10 CreERT2-mediated loss of GJB2, the AAV-mediated gene transfer of GJB2 did not demonstrate threshold improvement and, in some animals, exacerbated hearing loss and resulted in hair cell loss [20]. (6) The study by Yu et al. reported that, despite extensive virally expressed connexin 26 in cells lining the scala media, and intercellular gap junction network restoration in the organ of Corti of mutant mouse cochlea after the inoculation of modified AAV vectors, auditory brainstem responses did not show significant hearing improvement [21]. (7) The study by Maeda et al., although it shows that it may be possible to decrease the expression of the mutant allele and modify the deafness phenotype by RNAi protecting against hearing loss, was excluded because it targets an autosomal dominant form of GJB2 hearing loss, in particular the R75W mutation, and because it tested a RNA interference gene therapy by not using an AAV vector [22]. (8) The study by Chien et al. was excluded because, although targeting an autosomal recessive non-syndromic form of SNHL, i.e., the whirler (whrnwi/wi) mouse, a naturally-occurring model of DFNB31, and delivering gene therapy through an AAV vector, it does not show improved hearing sensitivity at any frequency according to ABR measurements in whirler mice that received whirlin gene therapy despite the restoration of whirlin expression in hair cells in neonatal mice [23]. (9) The study by Nakanishi et al. was excluded because it doesn't deal with a gene therapy administered by an AAV vector; in particular, the authors generated a transgenic mouse line, Tg[PTmc1::Tmc2], in which a Tmc2 cDNA under the transcriptional control of the Tmc1 promoter is expressed in mature cochlear HCs [24]. We have therefore included a total of 20 studies: 17 from the databases and 3 from the register, respectively. A flow diagram is attached (see Figure 1). In total, 17 records present preclinical studies testing AAV gene therapies in animal models of genetic hearing loss indicating different levels of hearing restoration after the administration of AAV gene therapy; three studies from the register are clinical trials that are currently under development and are testing AAV gene therapies in humans. Among the in vivo preclinical studies obtained from the databases, testing AAV gene therapies in animal models and showing hearing recovery, four studies targeted biallelic OTOF mutations models [8,9,25,26], one study targeted a mouse model of GJB2 deletion (DFNB1) [27], one targeted a TMPRSS3 mouse model (DFNB8) [28], one targeted a mouse model of STRC hearing loss (DFNB16) [29], five targeted mouse models of TMC1 mutant mice (DFNB7) [11,30–33], one targeted a mouse model of ILDR1 (DFNB42) [34], one targeted a mouse model of CABP2 hearing loss (DFNB93) [35], one targeted a mouse model of SYNE4 (DFNB76) [10], one targeted a mouse model of TMHS (DFNB67) [36] and

one targeted a mouse model of MSRB3 (DFNB74) [37]. Among the clinical studies included from registers, three trials currently ongoing and testing AAV gene therapies in patients with biallelic OTOF gene mutations have been found [38–40]. The studies have been listed in summary Tables 1 and 2.

**Table 1.** AAV gene therapy preclinical studies in DFNB animal models showing hearing improvement.

| Animal Model | DFNB Targeted | AAV Treatment Reagent | Publication Year |
|---|---|---|---|
| Otof−/− mice [8] | DFNB9 | Dual AAV2/6-Otof4 | 2019 |
| Otof−/− mice [9] | DFNB9 | Dual AAV quadY-F-Otof1 | 2019 |
| Otof−/− mice [25] | DFNB9 | Fl-Otof-overload AAVs | 2021 |
| Otof−/− mice [26] | DFNB9 | Dual AAV-PHP.eB-hOTOF NT and CT | 2022 |
| Cx26fl/fl/P0-Cre [27] | DFNB1 | AAV5- Gjb2 | 2015 |
| Tmprss3A306T/A306T [28] | DFNB8 | AAV2-hTMPRSS3 | 2023 |
| Strc−/− mice [29] | DFNB16 | Dual AAV9-PHP.B-Strc | 2021 |
| Tmc1-p.N193I [11] | DFNB7 | AAV9-PHP.B-CB6-hTMC1-WPRE | 2022 |
| Tmc1−/− mice [30] | DFNB7 | AAV9-PHP.B | 2021 |
| Tmc1−/− mice [31] | DFNB7 | AAV2/1-Cba-Tmc1 and AAV2/1-Cba-Tmc2 | 2015 |
| TMC−/− mice [32] | DFNB7 | AAV2/Anc80L65-Cmv-Tmc1ex1-WPRE | 2019 |
| Tmc1Y182C/Y182C;Tmc2+/+ mice [33] | DFNB7 | AAV-Anc80L65 | 2020 |
| Ildr1w−/− mice [34] | DFNB42 | Dual AAV2.7m8/AAV8BP2 | 2023 |
| Cabp2−/− mice [35] | DFNB93 | AAV2/1 and AAV-PHP.eB | 2021 |
| Syne 4−/− mice [10] | DFNB76 | AAV9-PHP.B | 2021 |
| Lhfpl5−/− [36] | DFNB67 | exo-AAV1-HA-Lhfpl5 | 2017 |
| MsrB3−/−mice [37] | DFNB74 | rAAV2/1-MsrB3-GFP | 2016 |

**Table 2.** AAV gene therapy clinical studies currently under development, according to clinicaltrials.gov.

| Mutation Targeted | DFNB Targeted | AAV Treatment Reagent | Current Clinical Phase | NCT Number | Sponsor |
|---|---|---|---|---|---|
| Biallelic mutations in the OTOF gene | DFNB9 | AAVAnc80-hOTOF | Phase 1/2 | NCT05821959 | Akouos, Inc. |
| Biallelic mutations in the OTOF gene | DFNB9 | AAV based gene therapy (DB-OTO) | Phase 1/2 | NCT05788536 | Decibel Therapeutics |
| Biallelic mutations in the OTOF gene | DFNB9 | AAV OTOV101N+OTOV101C | Phase 1 | NCT05901480 | Otovia Therapeutics |

*3.1. OTOF (DFNB9)*

The most promising results about a genetic therapy for autosomal recessive congenital hearing loss are those associated with hearing impairments caused by OTOF mutations.

The OTOF gene, responsible for producing the otoferlin protein, is a prominent factor in non-syndromic recessive sensorineural hearing loss. Over 160 OTOF mutations have been documented, with most patients experiencing stable, severe to profound hearing loss before language acquisition. OTOF is also recognized as the primary cause of non-syndromic recessive auditory neuropathy spectrum disorder [41–43]. The expression of OTOF is primarily observed in cochlear inner hair cells, where it plays a crucial role in synaptic exocytosis at the ribbon synapse [44]. Specifically, the otoferlin protein is essential for glutamate release in the synapse of inner hair cells, and mutations in the OTOF gene lead to DFNB9 [45].

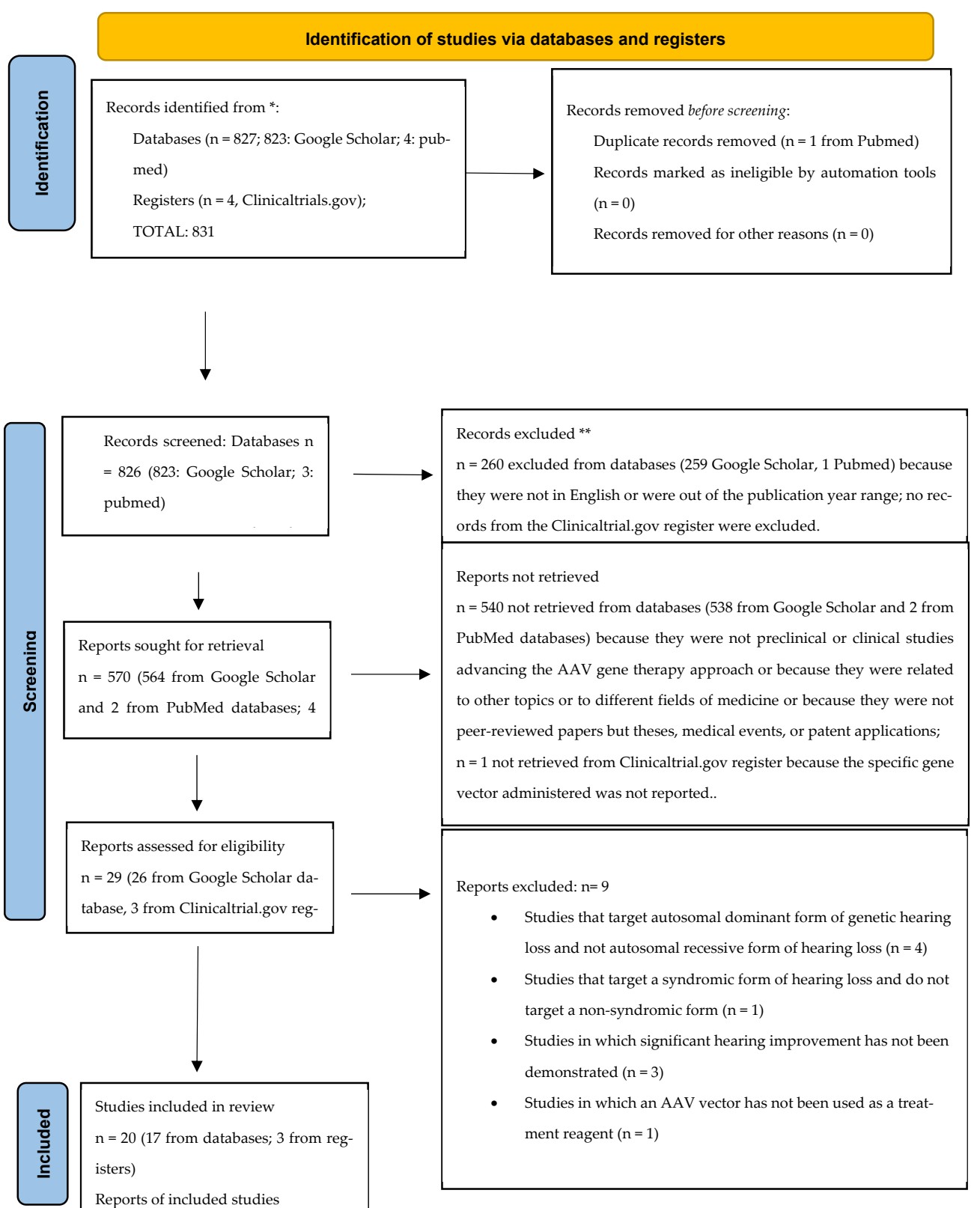

**Figure 1.** Flow diagram. * databases and registers (they are reported after the colon). ** all records excluded from databases and/or registers after the screening processes.

Four preclinical studies indicating hearing recovery through AAV gene therapy in OTOF mice knockout models have been found [8,9,25,26]. Delivery of the coding sequence

of some large genes, such as OTOF (~6 kb), is not straightforward because of the limited packaging capacity of standard AAV (<4.7 kb) [46]. This packaging issue has been tackled in preclinical gene replacement trials in DFNB9 mouse models using two approaches: dual-AAV and AAV overload.

The first model, dual-AAV, combines two different AAVs that carry 5′ and 3′ fragments of the coding sequence that recombine inside the IHC. Al-Moyed et al. demonstrated that reassembling the complete otoferlin cDNA in inner hair cells (IHCs) using a dual-rAAV strategy led to restored exocytosis and the partial recovery of auditory function in neonatal Otof−/− mice [8]. Similarly, Akil et al. achieved complete hearing restoration in both treated neonatal mice and young adult mice (P30) using the same dual-rAAV approach [9]. These findings provide optimism for future post-natal gene therapy trials in adult patients with DFNB9, as local gene therapy has shown the potential to rescue hearing even in young adult mice [47]. Rankovic et al. explored the AAV overload model by packaging the full-length OTOF coding sequence into a single AAV vector. The early postnatal injection of overloaded AAVs into the cochlea of OTOF-KO mice resulted in the specific expression of otoferlin in approximately 30% of all IHCs. This approach led to the partial restoration of hearing, as demonstrated by auditory brainstem response recordings and behavioral assays [25]. In another study, a novel dual-AAV-mediated gene therapy system based on protein trans-splicing was proposed. This system, after a single unilateral injection, effectively reversed bilateral deafness in OTOF−/− mice. The restoration of both hearing ability and synaptic vesicle release in inner hair cells was observed for at least 6 months [26].

Furthermore, we found three active clinical trials testing AAV gene therapy and recruiting patients diagnosed with hearing loss caused by biallelic OTOF gene mutations [38–40].

Another imminent clinical study from another biotech company, Sensorion, as reported by Qi et al. [48], should begin soon; however, this one has not been listed because it has not started yet, but a clinical trial application was submitted in the second quarter 2023.

To date, OTOF gene therapy programs are the only clinical trials testing an AAV gene therapy in human subjects.

### 3.2. GJB2 (DFNB1)

Non-syndromic autosomal recessive deafness exhibits genetic diversity, but a significant proportion of cases, up to 50%, can be attributed to a single locus on chromosome 13q11-12 known as DFNB1 [49]. The gene associated with DFNB1 is GJB2, which encodes for connexin 26, a gap junction protein responsible for creating channels between cells [50]. This protein plays a vital role in maintaining potassium homeostasis and facilitating intercellular signaling within the organ of Corti [51]. Other mutations have been identified over time, including those in GJB6 and GJB3, which encode connexin 30 and connexin 31, respectively; GJB2 remains the most common form of genetic autosomal recessive SNHL and the most common cause of hereditary deafness in many populations [52,53]. Clinically, it presents bilateral profound SNHL traditionally treated with cochlear implants.

Several preclinical in vitro and in vivo studies, like the one by Crispino et al., have shown that, by using AAV vectors, it is possible to restore connexin 26 protein expression and rescue gap junctions' function in the cochlea of GJB2 knockout mice [54]. In a study conducted by Yu et al., the restoration of a robust gap junction intercellular network among cochlear non-sensory cells was observed in vivo. This restoration was achieved through the inoculation of virally mediated gene therapy into the scala media of GJB2 knockout mice [21]. In another attempt to create a less severe GJB2 animal model, Guo et al. generated mice with the inducible Sox10iCreERT2-mediated loss of GJB2. These mice serve as a valuable tool for studying the role of connexin 26 in the cochlea and are useful for evaluating gene therapy vectors and developing therapies for GJB2-related deafness [20].

Despite the recent progress in preclinical studies showing the restoration of connexin 26 and the recovery of gap junctions' functions in GJB2 knockout mice, only one study by Takashi et al. showing signals of hearing improvement in knockout GJB2 animal models after AAV gene therapy administration was found [27]. A mouse model with

a specific gene deletion in the inner ear, targeting Connexin 26, was developed using a conditional approach controlled by the Protein 0 (P0) promoter. The study aimed to investigate the effectiveness and specificity of transcriptionally targeted AAV vectors for delivering Connexin 26 into the cochlea of GJB2-deficient neonatal mice. The results showed that AAV gene therapy successfully achieved the efficient expression of Connexin 26 in the non-sensory cells of the organ of Corti. Moreover, when the treatment was administered during the neonatal stage, it demonstrated the ability to prevent the progression of profound deafness, as evidenced by functional assessments of hearing. It seems that the time of intervention is crucial in the prevention of progression into profound deafness in GJB2 knockout mice; in the study by Takashi et al., the AAV gene therapy was administered in the neonatal period, while in the studies by Yu et al. and Guo et al., the AAV gene therapies were administered in postnatal conditional GJB2 knockout mice with no signals of hearing improvement, as shown by the auditory brainstem responses. Three separate studies utilizing conditional GJB2 knockout mouse models have demonstrated the critical involvement of Connexin 26 in the postnatal maturation and maintenance of the organ of Corti prior to the onset of hearing [55].

### 3.3. TMPRSS3 (DFNB8)

TMPRSS3, a member of the Type II Transmembrane Serine Protease family, is an enzyme primarily located in the neuron bodies of the spiral ganglion, the stria vascularis, and the epithelium of the organ of Corti. It possesses low-density lipoprotein receptor class A and scavenger receptor cysteine-rich domains [56]. While the specific function of TMPRSS3 remains unclear, it is believed to be involved in mechanoelectrical transduction by regulating the activity of ENaC, a sodium channel sensitive to amiloride. This regulation, in turn, affects the cochlear sodium concentration [57]. TMPRSS3 acts as a crucial factor for the survival and activation of cochlear hair cells during the onset of hearing [58]. Mutations in TMPRSS3 can result in two distinct phenotypes: prelingual (DFNB10) and delayed onset and postlingual (DFNB8) deafness [59]. To date, various mutations that contribute to the DFNB8/10 forms of deafness have been identified. While most affected individuals experience severe-to-profound hearing loss, the age of onset, severity, and rate of progression can vary, and no definitive correlation between the genotype and phenotype has been established [56]. Recently, a study demonstrated the rescue of hearing in aged mice with human DFNB8 deafness following a single administration of AAV-TMPRSS3 gene therapy [28]. To mimic the delayed-onset progressive hearing loss observed in human DFNB8 patients, a knockin mouse model with a common TMPRSS3 mutation associated with DFNB8 was generated. Using AAV2 as a vector to deliver the human TMPRSS3 gene, an injection of AAV2-hTMPRSS3 into the inner ear of adult knockin mice, at an average age of 18.5 months, resulted in the expression of TMPRSS3 in the hair cells and spiral ganglion neurons, leading to their survival. Furthermore, this single injection of AAV2-hTMPRSS3 resulted in the sustained rescue of auditory function, comparable to that of wild-type mice.

### 3.4. STRC (DFNB16)

The Stereocilin (STRC) gene, also known as DFNB16, is considered a significant contributor to bilateral mild-to-moderate sensorineural hearing loss [60,61]. Mutations in the STRC gene are the second most common cause of autosomal recessive non-syndromic deafness [62–65], and they are particularly prevalent in genetic hearing loss affecting sensory hair cells [66]. STRC gene expression is specific to sensory hair cells and is closely associated with the stereocilia, rigid microvilli that form the framework responsible for detecting sound stimuli [67]. Notably, stereocilin and otoancorin share similarities in their C-terminal sequences, suggesting their potential involvement in securing the tectorial membrane to the cellular structures of the organ of Corti. Functional stereocilin enables the cochlear amplifier to improve auditory thresholds by 60 dB, thereby enhancing sensitivity to faint sounds by a million-fold [68]. To develop gene therapy strategies for individuals with STRC-related hearing loss, Shubina-Oleinik et al. created a mouse model with a

targeted deletion in the STRC gene. They also designed a dual-vector protein recombination approach using AAV9-PHP.B capsids to replace the full-length wild-type STRC in the outer hair cells of DFNB16 mice [29]. Due to the large size of the STRC coding sequence (5430 bp), which cannot fit into a single AAV vector, a dual-vector strategy utilizing intein-mediated protein recombination was employed [69]. The vector injection was performed during the first postnatal week, resulting in the restoration of DPOAE and ABR thresholds in 50% of the treated mice. This restoration persisted for up to 12 weeks after injection, the latest time point tested. While STRC mutations lead to the loss of top connectors, disorganized hair bundle morphologies, and bundles detached from the tectorial membrane, this study revealed that STRC mutations did not cause early hair cell death, unlike other hearing loss mutations. These findings suggest that if a similar lack of hair cell death exists in humans with STRC mutations, there may be a broad temporal window for clinical intervention in individuals with DFNB16 [29].

*3.5. TMC1 (DFNB7)*

TMC1 plays a crucial role in auditory function in both mice and humans [70]. It functions as a mechanosensory transduction channel, converting mechanical stimulation from sound into electrical signals for hearing [71]. Studies have shown that TMC1 is expressed in the human fetal cochlea, in the inner and outer cochlear hair cells of mice, and in the neurosensory epithelia of the vestibular organs [72]. Deafness-causing mutations in TMC1 have been identified in humans, accounting for a significant percentage of inherited hearing loss cases [73]. These mutations can lead to severe-to-profound hearing loss, with one specific mutation, c.100C > T (p.R34X), being particularly common in autosomal recessive non-syndromic hearing loss [74,75]. Several preclinical studies have investigated the ability of AAV gene therapies to rescue hearing loss in mouse models with mutations in TMC1 (DFNB7 and DFNB11). For example, Marcovich et al. demonstrated the durable recovery of auditory function in mice injected with AAV9-PHP.B-CB6-hTMC1-WPRE, showing promising results in terms of auditory brainstem responses, distortion product otoacoustic emissions, cell survival, and biodistribution [30]. Another study by Wu et al. focused on using AAV9-PHP.B to deliver a Tmc1 gene replacement to different Tmc1-deficient mouse models in order to assess the extent of hearing recovery [31]. The researchers found that AAV9-PHP.B, when injected into the utricle, effectively transduced both inner hair cells and outer hair cells along the cochlea, resulting in the improved recovery of hearing and reduced cell death [31]. In terms of viral vectors, Askew et al. identified that AAV2/1 combined with the chicken β-actin promoter is an efficient combination for driving the expression of exogenous TMC1 in inner hair cells [32]. They demonstrated that exogenous TMC1 and TMC2 could restore sensory transduction, auditory brainstem responses, and acoustic startle reflexes in deaf mice. However, the effectiveness of AAV2/1 vectors in restoring cellular function was limited to inner hair cells and did not extend to outer hair cells [32]. To address this limitation, Nist-Lund et al. screened various viral capsids and discovered a synthetic AAV, Anc80L065, which efficiently transduced both inner and outer hair cells [11]. They observed significant improvements in auditory thresholds in mice injected with sAAV-Tmc1 using the Anc80L065 capsid compared to AAV2/1 vectors [11]. Yeh et al. investigated AAV gene therapy in a mouse model (Baringo mouse) with recessive hearing loss caused by a specific TMC1 mutation [33]. They utilized optimized cytosine base editors and guide RNAs to correct the pathogenic mutation in cultured Baringo-derived mouse cells and successfully restored sensory transduction in inner hair cells, preserved hair cell morphology, and partially rescued auditory function in vivo [33].

*3.6. ILDR1 (DFNB42)*

Epithelial cells possess specialized structures known as tight junctions (TJs) that contribute to the barrier function [76]. Several proteins, namely tricellulin, angulin-1/lipolysis-stimulated lipoprotein receptor (LSR), angulin-2/immunoglobulin-like domain containing receptor 1 (ILDR1), and angulin-3/immunoglobulin-like domain-containing receptor 2

(ILDR2), have been identified as components of TJs [77–80]. Mutations in TRIC (tricellulin) and ILDR1 (angulin-2/ILDR1) have been associated with autosomal recessive nonsyndromic deafness DFNB49 [81–83] and DFNB42 (MIM 609646) [84,85], respectively. ILDR1, an integral protein of the tricellular tight junction complex, is expressed in various cell types within the organ of Corti and the cochlear lateral wall of the inner ear. Tight junctions play a vital role in maintaining the proper ionic composition of the membranous compartments, including the +80-mV endocochlear potential that is necessary for sound mechanotransduction by cochlear hair cells [86–88]. Isgrig et al. conducted a study using synthetic adeno-associated viruses (AAVs) with different tropisms to deliver Ildr1 cDNA to the inner ear of Ildr1w−/− mice [34]. They employed AAV2.7m8 targeting the organ of Corti and AAV8BP2 targeting the cochlear lateral wall. The combined AAV2.7m8/AAV8BP2 gene therapy demonstrated improved cochlear structural integrity and auditory function in Ildr1w−/− mice, providing a novel approach for gene therapy in hereditary hearing loss cases involving diverse cell types within the cochlea [34].

### 3.7. CAPB2 (DFNB93)

The CAPB2 gene is responsible for encoding calcium-binding protein 2, which serves as a potent regulator of voltage-gated CaV1.3 channels in inner hair cells. Mutations in CAPB2 can lead to autosomal recessive moderate-to-severe non-syndromic hearing loss known as DFNB93 [89]. Audiological evaluations of affected individuals have revealed an auditory synaptopathy phenotype, often accompanied by intact otoacoustic emissions, indicating the preservation of outer hair cell function in some patients. Recent studies using a knockout mouse model have shed light on the underlying disease mechanism, suggesting the enhanced steady-state inactivation of CaV1.3 channels at the ribbon synapse of inner hair cells [90]. This results in a reduced number of available CaV1.3 channels for triggering exocytosis and supporting neurotransmission, leading to lower temporal precision in sound encoding and elevated hearing thresholds [91]. To address the hearing impairment associated with DFNB93, Oestreicher et al. explored the potential of AAV-mediated gene therapy [35]. They tested the ability of two AAV vectors, AAV2/1 and AAV-PHP.eB, to rescue hearing in mice lacking the CABP2 gene (CABP2−/−). Through injection into the round window membrane, efficient gene delivery and the partial restoration of hearing were achieved in the CABP2−/− mouse model of DFNB93 using both viral capsids. The AAV vectors demonstrated the successful transduction of inner hair cells, resulting in a partial improvement in hearing thresholds and increased ABR amplitudes in a significant proportion of treated animals. These findings indicate the restoration of the characteristic "non-inactivating" properties of voltage-gated CaV1.3 channels in inner hair cells [35].

### 3.8. SYNE4 (DFNB76)

Mutations in the SYNE4 gene, which encodes the protein nesprin-4, have been identified as the cause of autosomal recessive progressive high-frequency hearing loss [92,93]. Nesprins are localized to the outer nuclear membrane and interact with inner nuclear membrane SUN proteins as well as cytoskeletal elements in the cytoplasm [94]. Mouse models lacking SYNE4 or Sun1 have been shown to exhibit progressive hearing loss associated with DFNB76. In SYNE4 knockout mice (SYNE4−/−), hair cells develop normally, but the nuclei of outer hair cells gradually lose their basal position, leading to the subsequent degeneration of these cells [95]. To explore the potential of gene therapy in rescuing hearing loss caused by SYNE4 mutations, Taiber et al. utilized the SYNE4−/− mouse model and employed AAV9-PHP.B as a delivery vector [10]. AAV9-PHP.B, a synthetic adeno-associated virus, was used to deliver the coding sequence of SYNE4 to the inner ears of neonatal SYNE4−/− mice. Consistent with previous studies utilizing AAV9-PHP.B in the inner ear, the robust expression and transduction of both inner and outer hair cells along the cochlea were observed. The results demonstrated that the exogenous delivery of SYNE4 into neonatal SYNE4−/− hair cells led to the rescue of their morphology and to improved survival. The nuclei of injected outer hair cells were positioned at the basal part

of the cell, similar to wild-type mice, and the long-term survival of outer hair cells and the nuclei position were not significantly different. Furthermore, SYNE4 delivery resulted in the near-complete rescue of the auditory brainstem response (ABR) and distortion product otoacoustic emissions (DPOAE) thresholds. The best-performing animals achieved ABR thresholds as low as 15 dB for certain frequencies [10].

### 3.9. TMHS (DFNB67)

György et al. conducted a study using a mouse model with a targeted deletion of LHFPL5, also known as TMHS, to investigate the potential of AAV gene therapy [36]. LHFPL5 protein plays a crucial role in the mechanotransduction machinery of both outer hair cells (OHCs) and inner hair cells (IHCs). Its absence leads to the early degeneration of hair cells, profound deafness, and severe vestibular dysfunction [96]. In their study, György et al. explored the use of exosome-associated AAV vectors for delivering transgenes to cochlear hair cells and compared different injection routes in mice. Their findings demonstrated the efficiency of exosome-associated AAV vectors in the delivery of transgenes to cochlear and vestibular hair cells, both in vitro and in vivo, surpassing the performance of conventional AAV vectors in terms of gene transfer efficiency. To evaluate the efficacy of exosome-associated AAV vectors in treating TMHS-associated deafness, György et al. utilized a mouse-codon-optimized gene encoding LHFPL5 with a hemagglutinin (HA) tag at the N terminus, which was cloned into an AAV vector backbone under the CBA promoter. This exosome-associated AAV1-HA-Lhfpl5 construct was then injected into the cochlea of LHFPL−/− mice through the round window membrane. The study confirmed the functional mechanotransduction of exosome-associated AAV-transduced IHCs and OHCs, as well as the partial rescue of hearing, as demonstrated by auditory brainstem response (ABR) recordings at 4 weeks post injection [36].

### 3.10. MSRB3 (DFNB74)

Ahmed et al. discovered a correlation between the MSRB3 gene and human DFNB74, a genetic locus associated with autosomal recessive non-syndromic hearing loss [97,98]. They identified two mutations, p. Cys89Gly and p. Arg19X, in eight Pakistani families, demonstrating their impact on MSRB3 enzyme activity. Kim et al. conducted a study on an MSRB3 knockout mouse model (MSRB3−/−), which exhibited profound hearing loss similar to patients with MSRB3 mutations [37]. It is crucial to note that successful gene therapy for non-syndromic deafness with autosomal recessive inheritance requires the early expression of the delivered gene, considering most patients are deaf from birth. Bedrosian et al. demonstrated the safety and effectiveness of in utero gene transfer using recombinant adeno-associated virus (rAAV) vectors in the targeting of cochlear hair cell progenitors [99]. However, there is currently no evidence of the in vivo rescue of hearing function using in utero gene therapy in animal models with profound sensorineural hearing loss. Kim et al. administered rAAV expressing the MSRB3 gene to the embryonic otocyst of MsrB3−/− mice via trans-uterine microinjection, examining its effects on hair cell morphology and hearing [37]. The ABR thresholds were measured at 4 weeks, revealing that the treated ear exhibited waveforms resembling normal hearing thresholds, unlike the untreated ear. Despite its limitations, this study represents the first successful in utero rAAV-mediated gene therapy strategy able to rescue hair cells and restore hearing in a mouse model of congenital SNHL caused by a deafness gene.

### 4. Conclusions

In recent years, multiple studies have provided evidence that gene replacement therapy targeting the inner ear can effectively enhance auditory function in mouse models of hereditary hearing loss. As shown in the present review, different preclinical studies target specific DFNB genetic mutations causing autosomal recessive SNHL and demonstrate variable levels of hearing recovery. Several mutations of genes encoding different proteins with specific and complex rules in inner ear development and functions have been targeted

by gene therapy preclinical programs; for certain cases, such as the GJB2 gene, there is a need for a broader therapeutic window, and the potential of gene therapy to be directly clinically correlated and translated to humans may be limited. This is because in many forms of hereditary deafness, the degeneration of the sensory epithelium and neurons has likely already occurred during the early stages of development in the womb. As a result, targeting hair cells or spiral ganglion neurons postnatally may not yield significant therapeutic benefits. Gene therapy for sensorineural hearing loss (SNHL) has primarily focused on cochlear cells that are present after birth, and most gene delivery research in rodents is conducted on neonatal mice before hair cell degeneration occurs. However, promising results have also been achieved with gene delivery in adult transgenic mice, leading to the partial restoration of impaired hearing. This approach has shown success when performed prior to hair cell degeneration, as observed in OTOF mutations, where the cochlear structure remains preserved. We found, through the official clinicaltrials.gov register, three clinical studies currently testing AAV gene therapies in humans, specifically in patients diagnosed with sensorineural hearing loss caused by biallelic OTOF mutations. Gene therapy based on AAVs is rapidly becoming a new method for the treatment of hereditary deafness, but the road to complete and effective clinical translation for most mutations is still a long one.

**Author Contributions:** Conceptualization, D.B. and M.G.; methodology, D.B. and M.G.; validation, D.B., M.G.,C.D.F. and P.T.; formal analysis, D.B.; investigation, M.G.; resources, M.G.; data curation, D.B. and M.G.; writing—original draft preparation, M.G.; writing—review and editing, D.B. and M.G.; visualization, D.B. and M.G.; supervision, C.D.F. and P.T.; project administration, D.B. and M.G. All authors have read and agreed to the published version of the manuscript.

**Funding:** This research received no external funding.

**Conflicts of Interest:** There are no conflicts of interest, financial, or otherwise.

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
