# Peer review of "Autosomal Recessive Non-Syndromic Deafness: Is AAV Gene Therapy a Real Chance?"

_audiolres, doi:10.3390/audiolres14020022_

Round 1
Reviewer 1 Report
Comments and Suggestions for Authors
The article is of great interest.
Recommendations:
1. It would be better to shorten some detailed descriptions and to present the most important thing and author's opinion.
2. In the “Materials and Methods” section, the 2nd and 3rd sentences, in my opinion, are the same.
3. The detailed analysis include 20 articles, so it is better to place the material about the excluded 9 studies and a description of the Table 1 contents in the “Materials and Methods” section.
It is not necessary to list excluded articles using numbers.
4. Do not forget to make a new paragraph.
5. Is the genetherapy necessary in the case of the STRC gene deletions. As I know there is no evidence of outer hair cells death in this form. Patients have stable mild to moderate hearing loss, but not deafness.
6. Do not use only numbers instead of words, for example, 4 studies - 4, 1 article -1.
7. Writing gene symbols in italics.
8. Double-check the text for any misspellings and check the spelling of links, including Clinicaltrials.gov.
Author Response
Thank you very much for taking the time to review this manuscript. Please find the detailed responses below and the corresponding revisions changes in the re-submitted files.
1. It would be better to shorten some detailed descriptions and to present the most important thing and author's opinion. Response: We have shorten some detailed descriptions in the results section, and author's opinion have been deepened in the conclusions section
2. In the “Materials and Methods” section, the 2nd and 3rd sentences, in my opinion, are the same. Response: we have changed the 3rd sentence, the two sentences are now different
3. The detailed analysis include 20 articles, so it is better to place the material about the excluded 9 studies and a description of the Table 1 contents in the “Materials and Methods” section. It is not necessary to list excluded articles using numbers. Response: the material about the excluded 9 studies and a description of the Table 1 contents have been placed in the “Materials and Methods” section
4. Do not forget to make a new paragraph. Response: a new paragraph have been made before the correction reoprted in material and methods
5. Is the gene therapy necessary in the case of the STRC gene deletions. As I know there is no evidence of outer hair cells death in this form. Patients have stable mild to moderate hearing loss, but not deafness. Response: in respsonse to this comment, we have reported some parts of the text of Shubina-Oleinik O's article about STRC we listed in our review, in which rescue of hearing have been reported after an AAV gene therapy administration in a STRC mouse model:
"The mammalian inner ear has evolved a unique ability to amplify soft sounds, attenuate its response to loud sounds, and sharply tune its response in the frequency domain. Collectively, these functions, known as cochlear amplification, depend on the activity of sensory outer hair cells in the cochlea (1). Cochlear amplification requires cohesive outer hair cell bundles and a physical connection between hair bundle tips and the overlying tectorial membrane (2). A key protein required to couple hair bundles to the tectorial membrane and maintain cohesive hair bundles is known as stereocilin (3, 4). With functional stereocilin, the cochlear amplifier can improve auditory thresholds by ~60 dB, thus enhancing sensitivity to faint sounds by a million-fold. Unfortunately, the gene that encodes human stereocilin (STRC) is a common target of genetic mutations that cause recessive hearing loss, known as DFNB16 (5–7)."
"Some reports suggest that up to 16% of genetic hearing loss may be due to mutations in STRC, making DFNB16 the second most common form of genetic hearing loss and the most common form to affect sensory hair cells (5–8). Patients who carry STRC mutations lack cochlear amplification entirely and, as a result, suffer from reduced auditory sensitivity and have difficulty with frequency discrimination and speech perception (10). To address this significant unmet need, we determined the carrier frequency of STRC mutations, generated a mouse model of DFNB16, and designed a dual-vector protein-recombination strategy to replace full-length wild-type (WT) Strc in outer hair cells of DFNB16 mice that carry Strc mutations."
6. Do not use only numbers instead of words, for example, 4 studies - 4, 1 article -1. Response: request granted; we have used also words after numbers, in particular this modification have been made in the 3.0 results section
7. Writing gene symbols in italics. Response: All gene symbols have been reported in italics
8. Double-check the text for any misspellings and check the spelling of links, including Clinicaltrials.gov. Response: The text has been double-checked; also the spelling of links has been checked.
Reviewer 2 Report
Comments and Suggestions for Authors
Overall the information presented here about a AAV gene therapy for autosomal recessive non-syndromic deafness represents valuable information and written well. I can only suggest authors update the number of genes associated with autosomal recessive non-syndromic hearing loss to 78.
Author Response
Thank you very much for taking the time to review this manuscript. Please find the detailed responses below and the corresponding revisions changes in the re-submitted files.
Overall the information presented here about a AAV gene therapy for autosomal recessive non-syndromic deafness represents valuable information and written well. I can only suggest authors update the number of genes associated with autosomal recessive non-syndromic hearing loss to 78.
Response: Thank you very much for the compliments. We have updated the number of genes associated with autosomal recessive non-sindromic hearing loss to 78.
Reviewer 3 Report
Comments and Suggestions for Authors
The manuscript entitled “AUTOSOMAL RECESSIVE NON-SYNDROMIC DEAFNESS: IS AAV GENE THERAPY A REAL CHANCE?” is an important and timely review of the current state using AAV gene therapy to restore hearing in cases of autosomal recessive disorders. The following suggestions are made to the authors.
The abstract should clarify that the first sentence ( “The etiology of sensorineural hearing loss is heavily influenced by genetic mutations, with approximately 80% of cases attributed to genetic causes and only 20% to environmental factors.) refers to congenital hearing loss specifically and not sensorineural hearing loss more generally. Very likely other forms of sensorineural hearing loss (e.g., age-related hearing loss, noise-induced hearing loss, and ototoxicity) are more heavily influenced by environmental factors.
The advantages offered by the inner ear to gene therapy (The inner ear can be a good candidate for gene therapy for several reasons: 1) the cochlea is easily accessible by means of minimally invasive surgery; 2) it is reasonably self-contained anatomically, allowing easy and direct delivery of gene therapy; 3) the local application is performed within a relatively immune-protected environment 4) the organ is fluid-filled, thus allowing for widespread diffusion of the delivered gene.) are excellently summarized. I suggest that the authors provide references to support each of the four reasons.
The type of review performed by the authors is not per se clear and some additional methodological aspects would be a great addition to the research. First, the authors should ideally assess the risk of bias (using the Cochrane risk-of-bias tool) for the included studies, Second, inclusion of a meta-analysis on the effectiveness of studies (in cases where there are sufficient studies) would be an excellent addition. At the very least, the authors should consider including in the discussion the reasons for not assessing risk of bias and performing meta-analyses.
The conclusions would be much more valuable if there was a richer and more synthetic description of the key obstacles preventing advancement of gene therapy for the treatment of some forms of congenital hearing loss. As a start the authors should consider further explaining and developing key issues they have already excellently identified, including: 1) complex roles of these genes in inner ear development and functions; 2) need for a broader therapeutic window; and 3) need for more direct clinical correlation and translation to humans. Note also that in the above text “rules” should be “roles”.
Finally, the manuscript is generally well written but there are places where the English language needs to be refined. Contractions should be avoided. Text in the manuscript (especially in the first paragraph of Results section) and in Figure 1 describing the rationale for excluding studies should be properly constructed. (For example, “A total of 570 reports were sought for retrieval (564 from Google Scholar and 2 from PubMed databases; 4 from Clinicaltrials.gov): 540 reports from databases (538 Google Scholar, 2 PubMed) were removed because not preclinical or clinical studies advancing AAV gene therapy approach or because related to other kind of topic or to different field of medicine or because were not peer reviewed papers but thesis, medical events, or patent applications (538 from Google Scholar, 2 from PubMed); 1 report from Crinicaltrials.gov was removed because, despite a clinical study testing a gene therapy for OTOF gene hearing impairment, the specific gene vector administered was not reported.” Could/should be “A total of 570 reports were sought for retrieval: 564 from Google Scholar, 2 from PubMed databases, and 4 from Clinicaltrials.gov. Out of these, 540 reports from databases were removed—538 from Google Scholar and 2 from PubMed—because they were not related to preclinical or clinical studies advancing AAV gene therapy approaches. Other reasons for removal included reports on unrelated topics or different medical fields, or because they were non-peer-reviewed materials such as theses, medical event summaries, or patent applications. Additionally, 1 report from Clinicaltrials.gov was excluded because, although it was a clinical study testing gene therapy for OTOF gene-related hearing impairment, the specific gene vector administered was not disclosed.”
Comments on the Quality of English LanguagePlease see above.
Author Response
Thank you very much for taking the time to review this manuscript. Please find the detailed responses below and the corresponding revisions/corrections highlighted/in track changes in the re-submitted files
1) The abstract should clarify that the first sentence (“The etiology of sensorineural hearing loss is heavily influenced by genetic mutations, with approximately 80% of cases attributed to genetic causes and only 20% to environmental factors.) refers to congenital hearing loss specifically and not sensorineural hearing loss more generally. Very likely other forms of sensorineural hearing loss (e.g., age-related hearing loss, noise-induced hearing loss, and ototoxicity) are more heavily influenced by environmental factors. Response: The sentence have been clarified; we replaced "sensorineural hearing loss" with "congenital hearing loss".
2) The advantages offered by the inner ear to gene therapy (The inner ear can be a good candidate for gene therapy for several reasons: 1) the cochlea is easily accessible by means of minimally invasive surgery; 2) it is reasonably self-contained anatomically, allowing easy and direct delivery of gene therapy; 3) the local application is performed within a relatively immune-protected environment 4) the organ is fluid-filled, thus allowing for widespread diffusion of the delivered gene.) are excellently summarized. I suggest that the authors provide references to support each of the four reasons. Response: we decided to remove this part in which, for each point, the relevant source had not been reported; other authors have in fact asked us to cut some parts of the text in order to reduce it and make reading more fluent. Furthermore, by cutting this part, the previous and next part join together correctly
3) The type of review performed by the authors is not per se clear and some additional methodological aspects would be a great addition to the research. First, the authors should ideally assess the risk of bias (using the Cochrane risk-of-bias tool) for the included studies, Second, inclusion of a meta-analysis on the effectiveness of studies (in cases where there are sufficient studies) would be an excellent addition. At the very least, the authors should consider including in the discussion the reasons for not assessing risk of bias and performing meta-analyses. Response: we reported in the discussion the reasons for not assessing risk of bias and performing meta-analyses on effectiveness of studies; in particular, both the research about the preclinical and the clinical studies have been performed very rigorously using highly restrictive keywords in order to detect from the literature those preclinical studies that have reported signals of hearing improvement following the administration of a gene therapy via AAV (in fact, the literature is not particularly extensive regarding this type of studies about autosomal recessive non syndromic hearing loss, even more so in cases in which hearing improvement has been found following the administration of a gene therapy) and from the registers those clinical studies currently in development that are testing gene therapies using AAV in subjects with ARNSHL.
4)The conclusions would be much more valuable if there was a richer and more synthetic description of the key obstacles preventing advancement of gene therapy for the treatment of some forms of congenital hearing loss. As a start the authors should consider further explaining and developing key issues they have already excellently identified, including: 1) complex roles of these genes in inner ear development and functions; 2) need for a broader therapeutic window; and 3) need for more direct clinical correlation and translation to humans. Note also that in the above text “rules” should be “roles”. Response: ; we reported in the conclusion these sentences regarding the key obstacles preventing the advancement of gene therapy for the treatment of some forms of congenital hearing loss: "the complexity of genetic mutations leading to different forms of hearing loss make the underline histopathology particularly heterogeneous: this makes it necessary to rigorously define the exact site of lesions both in to the cochlea and in the cochleovestibular nerve before experimenting with any therapeutic approaches (including AAV gene therapies); this issue, both with the need for more direct clinical correlation and translation to humans, are to date key obstacles to preventing advancement of gene therapy for the treatment of some forms of congenital hearing loss."
English language refined.
Reviewer 4 Report
Comments and Suggestions for Authors
This review gives timely discussion about the progress of AAV-mediated gene therapy in autosomal recessive hearing loss.
Major points:
1. Abstract: “80% of cases attributed to genetic causes and only 20% to environmental factors”. There is no further discussion of this and no reference was provided. I might remove this or discuss it more in the discussion.
2. Introduction: AAV9-PHP.B transduces hair cells at high rates. As a “star vector”, if AAV9-PHP.B provides the higher transduction rate compared to any others, I would suggest to spell out. This is because “high rate” is ambiguous – for example, is transduction rate ~40 % in OHCs at the apex a high rate?
3. Result: There is a nice list of papers excluded from the study with reasons given. Is there a reason why the following study was not mentioned? AAV-mediated rescue of Eps8 expression in vivo restores hair-cell function in a mouse model of recessive deafness. Jeng et al., 2022. DOI:https://doi.org/10.1016/j.omtm.2022.07.012.
4. Table 1. The authors from reference 29 have corrected the AAV stereotype from 5 to 1.
5. Result 3.5. Askew et al. demonstrated that exogenous TMC1 “or” TMC2 could restore sensory transduction. They did not inject both TMC1 and TMC2 at the same time.
6. Result 3.8. The last 2 sentences need to be more accurate. Taiber et al provide statistics showing that, although hearing was rescued after treatment, there is a significant difference between treated and untreated animals in turns of ABR thresholds at some frequencies. Therefore, it is more appropriate to describe it as a substantial rescue instead of near-complete rescue.
Minor points:
1. Please double check the print: CaV1.3, the “V” in CaV1.3 should be subscript. The “w-/-“ in the Ildr1w-/- should be superscript. Same applies to other genes.
2. Result 3.4. STRC mutations did not cause early hair cell death, unlike “other” hearing loss mutations. I would suggest to change “others” to “most recessive”.
3. Spell out SNHL when used the first time in the article.
4. Table 1. It would be helpful if includes the age of the experiments conducted. This is because the authors made a good discussion on how age affects the effectiveness of gene therapy in the text.
Author Response
Thank you very much for taking the time to review this manuscript. Please find the detailed responses below and the corresponding revisions highlighted inthe re-submitted files.
Major points:
-
Abstract: “80% of cases attributed to genetic causes and only 20% to environmental factors”. There is no further discussion of this and no reference was provided. I might remove this or discuss it more in the discussion. Response: this is a general and well recognized sentence about the prevalence of sensorineural hearing loss at birth (as requested by another reviewer we modified the sentence changing the general "sensorineural hearing loss" with congenital hearing loss); we used this sentence to briefly emphasize in the abstract how prevalent genetics is in hearing loss, and therefore how gene therapy represents a new frontier in the treatment of deafness.
2. Introduction: AAV9-PHP.B transduces hair cells at high rates. As a “star vector”, if AAV9-PHP.B provides the higher transduction rate compared to any others, I would suggest to spell out. This is because “high rate” is ambiguous – for example, is transduction rate ~40 % in OHCs at the apex a high rate?. Response: we reported the exact tranduction rate as reported by Ivanchenko et al. in a preclinical testing for transgene expression "AAV9-PHP.B transduces nearly 100% of both IHCs and OHCs from base to apex at the higher doses (3.5 × 1011 and 7 × 1011 vector genomes)"
3. Result: There is a nice list of papers excluded from the study with reasons given. Is there a reason why the following study was not mentioned? AAV-mediated rescue of Eps8 expression in vivo restores hair-cell function in a mouse model of recessive deafness. Jeng et al., 2022. DOI:https://doi.org/10.1016/j.omtm.2022.07.012. Response: thank you for the advice; we have updated the list of the papers excluded; the preclinical study mentioned does not report any hearing improvement in the animals tested, so it is useful to cite why it has been excluded, so we have reported it.
4) Table 1. The authors from reference 29 have corrected the AAV stereotype from 5 to 1. Response: i dont' understand if it is necessary to make some corrections; for each study, we listed the exact AAV stereotype reported.
5. Result 3.5. Askew et al. demonstrated that exogenous TMC1 “or” TMC2 could restore sensory transduction. They did not inject both TMC1 and TMC2 at the same time. Response: thank you for the correction; we have changed the sentence
6. Result 3.8. The last 2 sentences need to be more accurate. Taiber et al provide statistics showing that, although hearing was rescued after treatment, there is a significant difference between treated and untreated animals in turns of ABR thresholds at some frequencies. Therefore, it is more appropriate to describe it as a substantial rescue instead of near-complete rescue. Response: thank you for the advice; we have changed the 2 sentences as requested.
Minor points:
-
Please double check the print: CaV1.3, the “V” in CaV1.3 should be subscript. The “w-/-“ in the Ildr1w-/-should be superscript. Same applies to other genes. Response: we have modified the text as requested
-
Result 3.4. STRC mutations did not cause early hair cell death, unlike “other” hearing loss mutations. I would suggest to change “others” to “most recessive”. Response: we have modified the text as requested
-
Spell out SNHL when used the first time in the article. Response: we have added it in the first sentence of the manuscript
-
Table 1. It would be helpful if includes the age of the experiments conducted. This is because the authors made a good discussion on how age affects the effectiveness of gene therapy in the text. Response: we have decided not to add the age in the table because our aim is to make a report of all the studies showing hearing improvement after AAV gene therapy and not a specific description of the features of each study. However, if the reviewer consider necessary this point, we will provide to add the age for every single study
-